# Evaluation of Piperacillin/Sulbactam, Piperacillin/Tazobactam and Cefoperazone/Sulbactam Dosages in Gram-Negative Bacterial Bloodstream Infections by Monte Carlo Simulation

**DOI:** 10.3390/antibiotics12020363

**Published:** 2023-02-09

**Authors:** Xueting Wang, Luying Xiong, Wei Yu, Chen Huang, Jinru Ji, Chaoqun Ying, Zhiying Liu, Yunbo Chen, Yonghong Xiao

**Affiliations:** 1State Key Laboratory of Infectious Disease Diagnosis and Treatment, National Clinical Medical Research Center for Infectious Diseases, Collaborative Innovation Center for Infectious Disease Diagnosis and Treatment, The First Hospital of Zhejiang University School of Medicine, Hangzhou 310000, China; 2Department of Respiratory Medicine, Ningbo Medical Center Lihuili Hospital, Ningbo 315000, China

**Keywords:** Monte Carlo simulation, bloodstream infections, pharmacokinetics/pharmacodynamics, piperacillin/sulbactam, piperacillin/tazobactam, cefoperazone/sulbactam

## Abstract

The optimal regimens of piperacillin/sulbactam (PIS 2:1), piperacillin/tazobactam (PTZ 8:1), and cefoperazone/sulbactam (CSL 2:1) are not well defined in patients based on renal function. This study was conducted to identify optimal regimens of BLBLIs in these patients. The antimicrobial sensitivity test was performed by a two-fold agar dilution method. Monte Carlo simulation (MCS) was used to simulate the probability of target attainment (PTA) and the cumulative fraction of response (CFR) for various dosing regimens in patients with different renal functions. For strains with an MIC ≤ 8/4 mg/L, PIS 4.5 g q6h achieved 99.03%PTA in the subset of patients with creatinine clearance (CrCL) > 90 mL/min. For patients with CrCL 60–90 mL/min, PIS 4.5 g q6h achieved 81.2% CFR; for those with CrCL 40–59 mL/min, PIS 4.5 g q8h achieved 80.25% CFR. However, for patients infected by ESBL-producing *Enterobacteriaceae*, PIS 4.5 g q6h achieved a CFR lower than 80%. For patients infected by *A. baumannii* with a CrCL of 31–60 mL/min, PIS 6.0 g q8h and 4.5 g q6h achieved 81.24% and 82.42% CFR, respectively. For those infected by *P. aeruginosa*, PIS 4.5 g q6h reached 90% CFR. PIS and PTZ achieved a similar CFR when piperacillin was at the same dose. The CFRs of CSL were much lower than those of the other two agents in *Enterobacteriaceae* and *P. aeruginosa* infections. The antibacterial spectrum of PIS is superior to that of PTZ and CSL. Higher dosages and dosing adjustment according to renal function should be considered to treat Gram-negative bacterial BSIs.

## 1. Introduction

The widespread use of antimicrobial drugs has increased the prevalence of antimicrobial resistance (AMR). In particular, infections caused by multidrug-resistant (MDR) Gram-negative bacteria have become a global threat to public health [1]. *Enterobacteriaceae* such as *Escherichia coli* and *Klebsiella pneumonia* are the main pathogens responsible for bloodstream infections (BSIs), urinary tract infections (UTIs), and hospital-acquired pneumonia; the resistance of these pathogens to antibiotics is driven mainly by the production of extended-spectrum β-lactamase (ESBL), which poses a great challenge to clinical management [2]. For nonfermentive bacteria such as *Acinetobacter baumannii* and *Pseudomonas aeruginosa*, β-lactamase production is likewise a major cause of AMR.

Piperacillin, a kind of semisynthetic penicillin, has broad-spectrum antibacterial activity but is unstable to β-lactamases. The yearly increase in the number of antimicrobial-resistant bacteria has reduced its efficacy. To overcome the resistance caused by β-lactamases, β-lactams and β-lactamase inhibitor combinations (BLBLIs) have been used widely in clinical practice. Piperacillin/sulbactam (PIS 2:1), piperacillin/tazobactam (PTZ 8:1), and cefoperazone/sulbactam (CSL 2:1) are the most widely used drugs in China. PIS, a combination of β-lactamase inhibitors developed in China, was launched in 2005 and showed good antibacterial activity against Gram-negative bacteria, including *Acinetobacter* spp.; this drug is used mainly to treat respiratory infections, UTIs, and intra-abdominal infections.

In the context of rapidly changing AMR and the serious lag in the development of new antimicrobial drugs, pharmacokinetic/pharmacodynamic (PK/PD) principles are often used to design and optimize antimicrobial drug-dosing regimens. For β-lactams, which show time-dependent activity, the %fT > minimal inhibitory concentration (MIC) is the best predictor of efficacy [3]. Monte Carlo simulation (MCS) is a statistical modeling approach based on PK parameters and antimicrobial susceptibility. It can predict the probability of success of different dosing regimens against target bacterial infections by simulating PK in a large number of virtual patients [4]. Based on PK parameters from a representative population, along with antimicrobial susceptibility data from clinical isolates, the program can calculate the probability of target attainment (PTA) and the cumulative fraction of response (CFR), which can be generated to guide the optimal dosing regimens for treating of infections. 

In this study, an MCS was used to compare the PTA and CFR in different regimens to optimize bloodstream infection therapy with PIS and its comparators.

## 2. Results

### 2.1. In Vitro Activity of Antibacterial Agents

The study included 5692 strains of *Enterobacteriaceae* (2510 strains of ESBL-positive isolates), 116 strains of *A. baumannii*, and 280 strains of *P. aeruginosa*. Three antibacterial drugs (PIS, PTZ, and CSL) showed potent in vitro activity against *Enterobacteriaceae*. MIC_90_ values of PIS, PTZ, and CSL against ESBL-negative *Enterobacteriaceae* were 16/8, 16/4, and 2/1 mg/L, respectively, and sensitivity rates were all higher than 80%. The activity against ESBL-producing *Enterobacteriaceae* was much weaker, with MIC_90_ values of 256/128, 128/4, and 128/64 mg/L, respectively. At the same time, the sensitivity rates of *A. baumannii* to PIS and CSL were 83.6% and 76.7%, respectively, while those of *P. aeruginosa* were all higher than 90% (Table 1).

### 2.2. Monte Carlo Simulation Analysis

#### 2.2.1. PTA

For strains with an MIC value of 8 mg/L (concentration of piperacillin), PIS administered as a 4.5 g q6h regimen achieved 99.03%PTA in patients with a creatinine clearance (CrCL) > 90 mL/min; PIS administered as 4.5 g q8h achieved 96.24%PTA and 99.91%PTA in patients with a CrCL of 60–90 mL/min and 40–59 mL/min, respectively; PIS 6.0 g q12h and 3.0 g q8h achieved 97.85% and 97.89%PTA in patients with CrCL of 20–39 mL/min, while in patients with a CrCL < 20 mL/min, PIS only dosing with a 3.0 g q12h regimen achieved 99.55%PTA (Figure 1). When PIS administered as 1.5 g q12h and 3.0 g q12h, only in patients with CrCL < 20 mL/min could PIS 3.0 g q12h reached 90%PTA for strains with an MIC of 8 mg/L (not showed in figure). When the daily dosage of piperacillin in PIS and PTZ was the same, a similar PTA was achieved for strains with the same MIC value (Figure 2). For patients with different renal function who were infected with strains with an MIC of 16 mg/L, CSL (even the high-dose regime of 4.5 g q8h) did not achieve 90%PTA (Figure 3). 

#### 2.2.2. CFR

All of the PIS dosing regimens (3.0 g q8h, 3.0 g q6h, 4.5 g q8h, 4.5 g q6h, and 6.0 g q8h, 6.0 g q12h) achieved a CFR lower than 80% in patients with a CrCL > 90 mL/min who were infected with *Enterobacteriaceae*. For patients with a CrCL of 60–90 mL/min, 4.5 g q6h achieved an 81.34% CFR; for patients with a CrCL of 40–59 mL/min, 4.5 g q8h achieved an 80.25% CFR; for patients with a CrCL of 20–39 mL/min, all PIS dosing regimens (except for 3.0 g q8h and 6.0 g q12h) achieved an 80% CFR; and for patients with a CrCL < 20 mL/min, the 4.5 g q6h and 6.0 g q8h regimens achieved an 91.81% and 91.8% CFR, respectively. Neither PIS 1.5 g q12h nor PIS 3.0 g q12h reached 80% CFR for patients with various renal functions (not shown in table). In patients infected with ESBL-producing *Enterobacteriaceae*, all dosing regimens achieved a low CFR, with a maximum of no more than 80%.

For patients infected with *A. baumannii* and a CrCL > 60 mL/min, all dosing regimens of PIS achieved a low CFR, with a maximum of no more than 50%. For patients with a CrCL of 31–60 mL/min, sulbactam administered as 6.0 g/d reached an 80% CFR; for patients with a CrCL of 10–30 mL/min or <10 mL/min, all dosing regimens reached an 80% CFR. When sulbactam was administered as at least 3.0 g/d, it achieved a 90% CFR in patients with a CrCL < 10 mL/min. When the same daily dose of sulbactam was administered along with PIS or CSL, the CFR achieved with PIS was higher in patients with a similar level of renal function.

For patients infected by *P. aeruginosa* with various levels of renal function, PIS administered as 4.5 g q6h and 6.0 g q8h reached a 90% CFR. When piperacillin was administered at the same dose in both PTZ and PIS, the CFR achieved with PTZ was similar to that of PIS in patients with the same level of renal function. In addition, the CFR achieved using various CSL-dosing regimens was lower than that achieved by the other two drugs.

For patients with a CrCL > 90 mL/min who were infected with Gram-negative bacteria, all dosing regimens of PIS achieved a CFR lower than 80%. For patients with a CrCL of 60–90 mL/min, PIS administered as 4.5 g q6h achieved an 81.34% CFR; for patients with a CrCL of 40–59 mL/min, PIS administered as 4.5 g q8h achieved an 80.25% CFR; for patients with a CrCL of 20–39 mL/min, all dosing regimens except 3.0 g q8h and 6.0 g q12h reached an 80% CFR. For patients with a CrCL < 20 mL/min, all dosing regimens reached an 80% CFR; PIS administered as 4.5 g q6h and 6.0 g q8h achieved a 92% and 90.22% CFR, respectively. For patients with a similar level of renal function, PTZ achieved a CFR similar to that of PIS when the piperacillin was at the same dose. The CFR achieved by CSL at each dosing regimen was lower than 80% in patients with differing renal function (Table 2).

## 3. Discussion

In recent years, the prevalence of ESBL-producing *Enterobacteriaceae* has posed a major challenge to the treatment of infections [2]. Carbapenems are the treatment of choice for serious infections due to ESBL-producing organisms; however, AMR is become increasingly prominent as their use increases [5]. To release the resistance pressure on bacteria caused by the overuse of carbapenems, BLBLIs have become an important option for the treatment of mild to moderate infections [6,7]. One of the main reasons for the controversy over the use of BLBLIs is that drug efficacy reduces as the inoculum increases, which is referred to as the inoculum effect. In addition, ESBL-expressing genes are usually located on plasmids, and often encode other resistance mechanisms simultaneously; furthermore, plasmid expression is increased upon exposure to antibiotics, making β-lactamase inhibitors much less effective [8]. Therefore, BLBLIs are recommended for therapy of mild to moderate infections by ESBL-producing bacteria in China [9].

In this study, an MCS was performed to evaluate the efficacy of three BLBLIs commonly used in clinical practice in China and to choose the optimal dosing regimens for the treatment of different levels of renal function. PIS is a BLBLI developed independently in China, with recommended doses of 1.5 g or 3.0 g q12h, and increased to 6.0 g q12h for the treatment of severe or refractory infections. However, the recommended dose of instruction is insufficient to achieve the target of %fT > MIC through MCS. Although we found that PIS showed potent in vitro activity against *Enterobacteriaceae*, for strains with an MIC ≤ 8 mg/L, only PIS administered at high doses (4.5 g q6h) reached 90%PTA in patients with CrCL values > 90 mL/min. This suggests that PIS is underdosed in the treatment of infection. Even PIS administered at high doses (4.5 g q6h) did not reach 80% CFR in patients with CrCL values > 90 mL/min who were infected by *Enterobacteriaceae*; a higher dose regimen may be needed, but drug toxicity also should be taken into account. The recommended dose should be 4.5 g q6h for patients with CrCL values of 60–90 mL/min, 4.5 g q8h for patients with a CrCL of 40–59 mL/min, 3.0 g q6h for patients with a CrCL of 20–39 mL/min, and 3.0 g q8h or 6.0 g q12h for those with CrCL < 20 mL/min. For patients with CrCL > 60 mL/min who are infected with *A. baumannii*, even if PIS administered as 4.5 g q6h also failed to achieve the target CFR. The recommended dose for those with CrCL of 31–60 mL/min is 6.0 g q8h, and 3.0 g q8h for those with CrCL of 10–30 and <10 mL/min. A multicenter study of PIS in 2004 found that similar clinical efficacy (91.55% vs. 91.18%, 90.27% vs. 100%, *p* > 0.05) and good safety were achieved (7.69% vs. 8.33%, *p* > 0.05) for PIS (4/1 g iv q8h) and PTZ (4/0.5 g iv q8h) in the treatment for respiratory tract infections and UTIs (no specific type of infection was stated) [10]. In 1997, a study on febrile neutropenia in pediatric cancer patients found that PIS was more efficient than PTZ, but the efficacy rate was only 34.3% [11]. With the increase in AMR, we expect that PIS will be effective for the treatment of Gram-negative bacterial BSIs if the dose and frequency of administration are increased.

For ESBL-producing *Enterobacteriaceae*, three drugs, even when administered at higher than the recommended doses, failed to achieve the target CFR. According to the 2022 guidelines of The Infectious Diseases Society of America (IDSA), BLBLIs are not recommended for BSIs caused by ESBL-producing *Enterobacteriaceae* even if they show potent activity in vitro [12]. According to this study, strains with an MIC > 8 mg/L for PTZ resulted in a low CFR when administered at the recommended dose. In addition, clinical studies showed that PTZ was less effective than carbapenems when used to treat BSIs caused by ESBL-producing *Enterobacteriaceae*, and the 30-day mortality rate of patients increased when the strain had an MIC > 16 mg/L for PTZ. Therefore, the use of PTZ is not recommended, especially when the MIC is > 16 mg/L [13].

When the daily dose of piperacillin in PIS and PTZ administered was the same, the PTA achieved by PTZ was similar to that of PIS for strains excluding *A. baumannii*. For patients with a CrCL ≤ 90 mL/min, the target CFR was achieved even using PTZ at low-dose regimen (2.25 g q6h and 3.375 g q8h), whereas for those with a CrCL > 90 mL/min, an increase to 3.375 g q6h was required to achieve the target CFR.

Sulbactam showed potent activity against *A. baumannii*, the same as other β-lactams, which mediated through inhibition of protein-binding proteins (PBPs), including PBP 1a/b and PBP3 [14].Our study found that PIS, PTZ and CSL all showed excellent antibacterial activity against *A. baumannii* with susceptible rates higher than 90%,as previous studies found. Sulbactam exhibited time-dependent bactericidal activity in previous study, similar to β-lactams. In addition, the %fT > MIC in the thigh/lung infection models (R^2^ = 0.95/0.96, respectively) appeared to be the most predictive of in vivo effects [15]. An in vivo pharmacodynamic study found that bactericidal activity of sulbactam against *A. baumannii* in a murine thigh and lung infection model was observed when the %fT > MIC targets were approximately 40% and 30% [15]. To better predict bacterial killing, we selected a target ≥40% for sulbactam in our study.The achieved PTA and CFR were higher than those for PTZ, in accordance with the results of antimicrobial susceptibility testing. Sulbactam is considered one of the most effective concomitant medications when used together with other effective antibiotics for the treatment of these pathogens. According to foreign recommendations, the routine dose of sulbactam is no more than 4.0 g/d for mild to moderate *A. baumannii* infections, while for severe infections, the dose of sulbactam could be increased to 6.0 to 8.0 g/d [16]; these data are consistent with the results of our study.

PIS and PTZ showed potent activity against *P. aeruginosa* strains, with an MIC ≤8 mg/L. PIS administered at 4.5 g q6h reached 90%PTA, and all simulated administered doses reached 80% CFR.

The PD of β-lactams have been well described and applied to the design of dosing regimens to optimize drug efficacy in recent years. For BLIs, rigorous study is a recent undertaking; limited studies evaluated the %fT > threshold was the PK/PD index best associated with efficacy for tazobactam. The target tazobactam concentration threshold changed with the amount of β-lactmase transcription when administered in combination and was 2 mg/L for the high-level CTX-M-15- or TEM-1-producing strains combined with piperacillin. When given in combination with piperacillin for CTX-M-15- or TEM-1-producing strains, associated with 1 − log_10_ CFU/mL reductions from baseline at 24 h were approximately 56% and 63%, respectively [17]. The CT has not been conclusively determined yet and likely needs to incorporate enzyme expression [17]. However, early studies determined that the use of this fixed concentration of tazobactam achieved the goal of BLIs to suppress β-lactamase activity [18]. Furthermore, studies showed that current doses of PTZ provide sufficient concentrations of tazobactam to allow piperacillin to achieve its PD target of ≥40–50%fT > MIC [19].

This study has some limitations. Firstly, the PTZ dosage used in the MCS was at a proportion of 8:1, whereas in the antimicrobial sensitivity test it was not (tazobactam at a fixed concentration of 4 mg/L). This may lead to a bias in the %fT > MIC. Secondly, in severe infections, the PK of antimicrobial drugs can be affected by the altered pathophysiological conditions of the patients; therefore, using PK parameters from healthy people in the study may lead to bias, which means the results may not fully reflect the situation in actual patients. Thirdly, due to the limited data, the CrCL ranges used for PK parameters of cefoperazone and sulbactam were not the same as piperacillin, which may have some influence on the result analysis. Furthermore, there is no exact correspondence between the in vitro MCS predictions and clinical efficacy; the results presented herein need to be validated in clinical trials.

## 4. Materials and Methods

### 4.1. Bacterial Isolates

In 2019, the Blood Bacterial Resistance Investigation Collaborative System (BRICS) collected bacterial strains from blood cultures obtained in 54 hospitals in different regions of China; test strains included non-carbapenem-resistant *Enterobacteriaceae*, *A. baumannii*, and *P. aeruginosa*. Pathogens were isolated using the API20 system in accordance with the clinical microbiological methods and identified by matrix-assisted laser desorption ionization–time-of-flight mass spectrometry.

### 4.2. Antimicrobial Agents and Culture Media

Piperacillin (potency 95%, lot no. P1200717), sulbactam (potency 91%, lot no. 03191203), and tazobactam (potency 95%, lot no. 03191203) were purchased from Suzhou Erye Pharmaceutical Co. Cefoperazone (potency 99%, lot no. 130420-201105) was purchased from the National Institutes for Food and Drug Control. Mueller–Hinton agar (MHA) was purchased from OXOID (UK, lot no. 2989738).

### 4.3. Antimicrobial Susceptibility Testing 

The MICs of PIS, PTZ, and CSL were determined at our laboratory by the agar dilution method according to Clinical and Laboratory Standards Institute (CLSI) guidelines (Thirty-Second Informational Supplement: M100-S32). PIS and CSL were prepared at a ratio of 2:1, and tazobactam in PTZ was used at a fixed concentration of 4 mg/L. Plates were incubated at 35 °C for 16–20 h. *E. coli* ATCC 25922 and *P. aeruginosa* ATCC 27853 were the quality-control strains.

CLSI criteria were used to interpret the results according to the interpretive standards for PTZ (*Enterobacteriaceae*: ≤8/4 mg/L = sensitive, ≥32/4 mg/L = resistant; *P. aeruginosa* and *A. baumannii*: ≤16/4 mg/L = sensitive, ≥128/4 mg/L = resistant), and PIS (*Enterobacteriaceae*: ≤8/4 mg/L = sensitive, ≥32/16 mg/L = resistant; *P. aeruginosa*: ≤16/8 mg/L = sensitive, ≥128/64 mg/L = resistant) [5]. The FDA criteria for cefoperazone was used to interpret the results for CSL (*Enterobacteriaceae* and *P. aeruginosa*: ≤16/8 mg/L = sensitive, ≥64/32 mg/L = resistant). For PIS and CSL against *A. baumannii*, the CLSI criteria for sulbactam within ampicillin/sulbactam (≤4 mg/L = sensitive and ≥16 mg/L = resistant) were used [20].

### 4.4. Pharmacokinetics (PK) Parameters Used for Simulations

Two previously published one-compartment PK models for PTZ and CSL were used, derived from 50 participants (8 healthy volunteers, 42 patients with various degrees of renal failure) and 24 participants (6 healthy volunteers, 18 patients with various degrees of renal failure), respectively. The dosing regimens for PTZ and CSL were single intravenous doses of 3.0/0.375 g and 2.0/1.0 g, respectively. Blood samples were collected in good time to determine the drug serum concentrations, and adequate PK analysis with appropriate parameters were performed (Table 3) [21,22]. The PK parameters of PIS referred to that of the above two drugs.

### 4.5. Monte Carlo Simulation

PIS, PTZ, and CSL show time-dependent bactericidal effects, which correlate with the best %fT > MIC. %fT > MIC was calculated using the following one-compartment intravenous infusion equation, where Ln is the natural logarithm, dose is the intermittent dose in mg, f is the fraction of unbound drug, and DI is the dosing interval in hours:(1)$$\%\mathrm{fT}\;>\;\mathrm{MIC}=\ln\;(\frac{\mathrm{Dose}\times f}{\mathrm{Vd}\times \mathrm{MIC}})\times \frac{\mathrm{Vd}}{\mathrm{CL}}\times \frac{100}{\mathrm{DI}}$$

MCS (Oracle Crystal ball, version 11.1.2.4.400) was performed to analyze the PTA and CFR of PIS, PTZ, and CSL in different regimens based on CrCL. The PK/PD target value for piperacillin and sulbactam were both ≥40 %fT > MIC, whereas that for cefoperazone was ≥50 %fT > MIC [8]. The dosing regimen was considered optimal if it provided ≥90%PTA. The CFR can be calculated from the following formula:(2)$$\mathrm{CFR}\;=\sum_{i=1}^{n}\mathrm{PTAi}\times \mathrm{Fi}$$

PTAi is the probability of achieving the target value at a given MIC value, and Fi is the relative probability of the distribution of each MIC value in the strains. In this study, dosing regimens were considered optimal if they provided ≥80% CFR.

## 5. Conclusions

PIS shows good antibacterial activity against *Enterobacteriaceae*, *P. aeruginosa*, and *A. baumannii* compared with PTZ and CSL. The study suggests that the recommended dose of PIS in the current prescribing insert is insufficient; therefore, increasing the dosage and frequency of administration should be considered for the treatment of BSIs, as well as dose adjustment based on renal function. A dosing regimen of 4.5 g q6h for PIS can be used for strains with MIC < 8 mg/L, as well as for the empirical treatment of mild to moderate Gram-negative bacterial BSIs. The PTA of PTZ and PIS were similar when administrated at the same piperacillin dose; the CSL-simulated dosing also suggested prominent underdosing for Gram-negative bacterial BSIs.

## Figures and Tables

**Figure 1 antibiotics-12-00363-f001:**
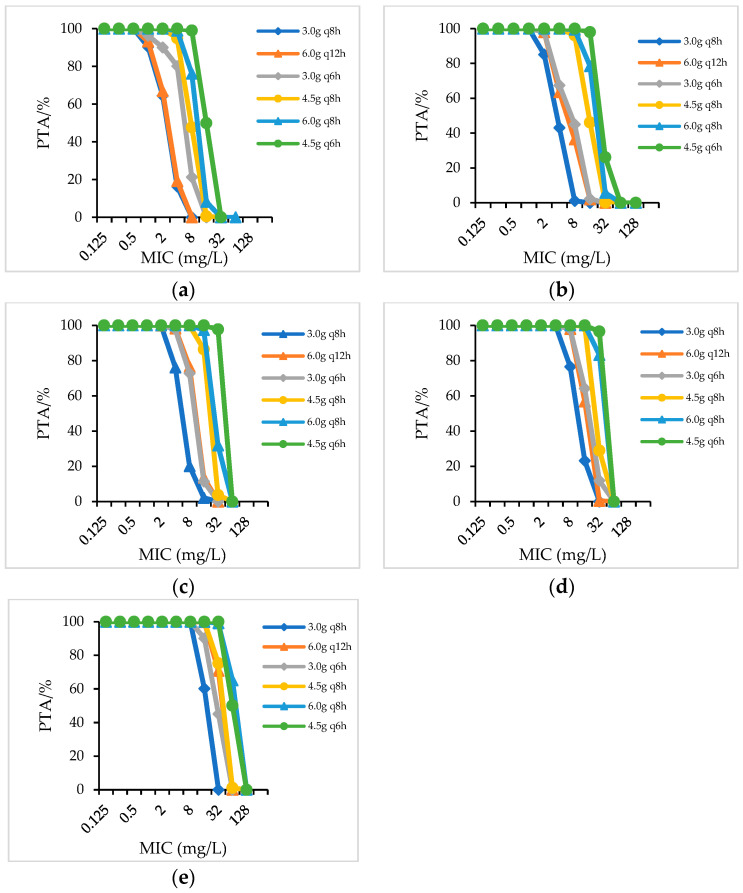
PTA for PIS in patients with different renal functions. (**a**) CrCL > 90 mL/min; (**b**) CrCL 60–90 mL/min; (**c**) CrCL 40–59 mL/min; (**d**) CrCL 20–39 mL/min; (**e**) CrCL < 20 mL/min.

**Figure 2 antibiotics-12-00363-f002:**
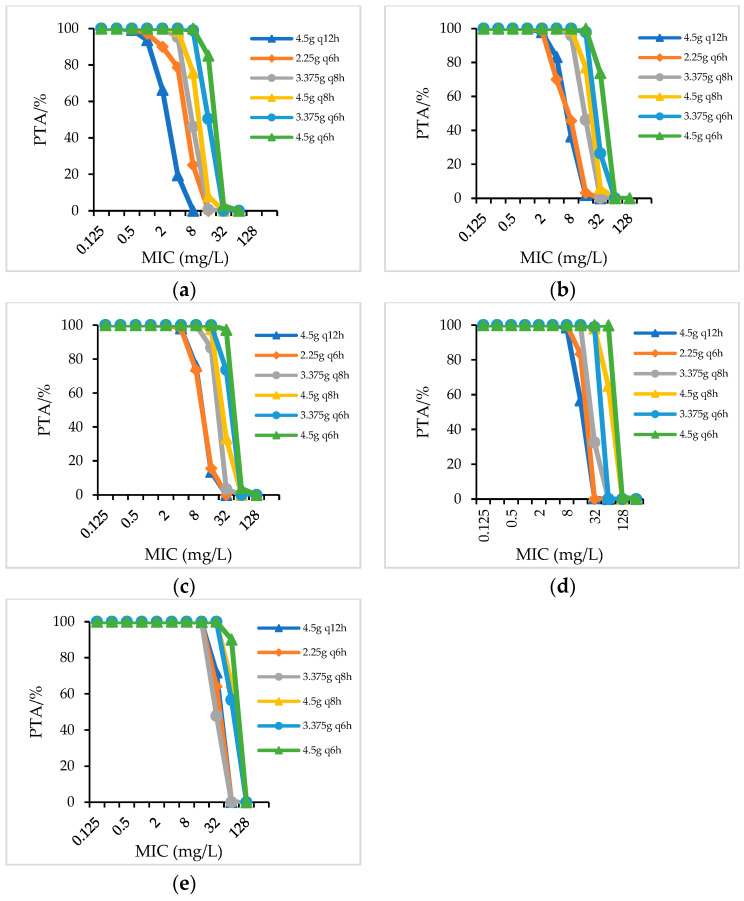
PTA for PTZ in patients with different renal functions. (**a**) CrCL > 90 mL/min; (**b**) CrCL 60–90 mL/min; (**c**) CrCL 40–59 mL/min; (**d**) CrCL 20–39 mL/min; (**e**) CrCL < 20 mL/min.

**Figure 3 antibiotics-12-00363-f003:**
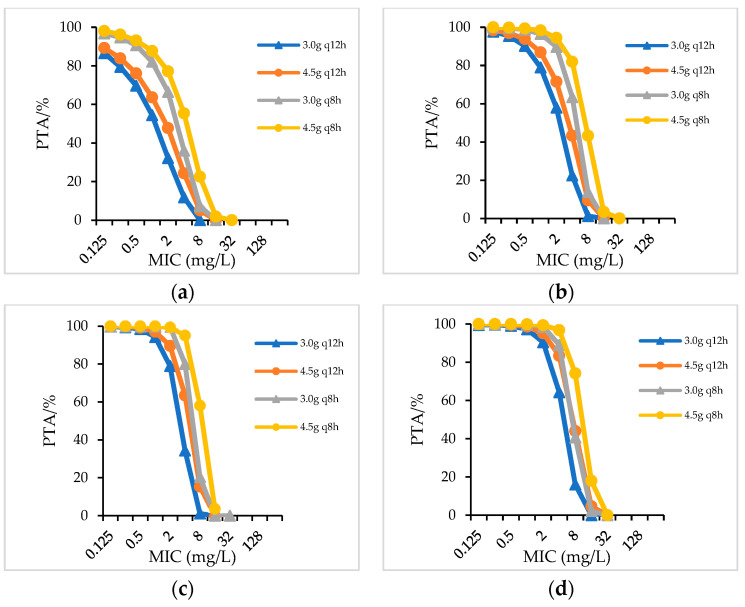
PTA for CSL in patients with different renal functions. (**a**) CrCL > 60 mL/min; (**b**) CrCL 31–60 mL/min; (**c**) CrCL 10–30 mL/min; (**d**) CrCL < 10 mL/min.

**Table 1 antibiotics-12-00363-t001:** In vitro antibacterial activity of PIS, PTZ, and CSL against Gram-negative isolates from blood culture.

Organism, Phenotype/Genotype (No. of Isolates)	Antimicrobial Agent	MIC (mg/L)			% Susceptible	% Resistant
		MIC Range	MIC_50_	MIC_90_		
*Enterobacteriaceae* (5692)	PIS	0.5/0.25–56/128	8/4	64/32	62.9	20.9
	PTZ	0.125/4–256/4	4/4	64/4	76.2	16.2
	CSL	0.125/0.06–128/64	1/0.5	32/16	85.9	7.0
*Enterobacteriaceae*, ESBL+ (2510)	PIS	0.5/0.25–256/128	16/8	256/128	30.52	46.1
	PTZ	0.125/4–256/4	8/4	128/4	54.15	37.81
	CSL	0.25/0.125–128/64	16/8	128/64	59.53	26.54
*Enterobacteriaceae*, ESBL− (3182)	PIS	0.5/0.25–256/128	4/2	16/8	81.64	9.81
	PTZ	0.125/4–128/4	4/4	16/4	83.87	8.85
	CSL	0.25/0.125–128/64	0.25/0.125	2/1	97.02	2.2
*A. baumannii* (116)	PIS	0.5/0.25–256/128	2/1	16/8	83.6	9.4
	PTZ	1/4–128/4	4/4	128/4	56.8	33.6
	CSL	1/0.5–128/64	4/2	16/8	76.7	7.7
*P. aeruginosa* (280)	PIS	0.5/0.25–256/128	2/1	16/8	93.9	2.5
	PTZ	0.25/4–128/4	2/4	4/4	95.0	3.9
	CSL	4/2–128/64	4/2	16/8	91.8	1.1

PTZ, piperacillin/tazobactam; PIS, piperacillin/sulbactam; CSL, cefoperazone/sulbactam; ESBL, extended-spectrum β-lactamases.

**Table 2 antibiotics-12-00363-t002:** CFRs of the different dosage regimens of PIS, PTZ, and CSL against Gram-negative isolates from blood cultures.

Species/CrCL(mL/min)		Dosage Regimens (CFR/%)															
		PIS						PTZ						CSL ^a^			
		3.0 g q8h	6.0 g q12h	3.0 g q6h	4.5 g q8h	6.0 g q8h	4.5 g q6h	4.5 gq12h	3.375 g q8h	2.25 g q6h	4.5 g q8h	3.375 g q6h	4.5 g q6h	3.0 gq12h	4.5 gq12h	3.0 g q8h	4.5 g q8h
Gram-negative bacteria	>90	58.39	35.43	72.61	64.21	76.18	78.71	67.87	69.38	76.18	77.96	80.19	82.56	-	-	-	-
	60–90	61.67	68.75	76.42	79.7	76	81.34	78.99	80.09	81.47	83	85.11	87	40.21	46.19	53.77	58.77
	40–59	68.92	72.11	79.26	80.25	81.82	86.24	81.23	83.54	83.78	85.65	87.25	87.49	51.03	57	62.01	68.65
	20–39	78.68	79.87	82.41	81.95	87.42	88.67	83.25	84.86	84.97	87.22	87.99	89.09	56.63	61.68	65.49	70.84
	<20	84.77	85.71	89.34	88.49	90.22	92	86.85	88.23	87.69	90.1	90.35	91.78	62.03	68.27	68.5	75.96
*Enterobacteriaceae*	>90	50.99	55.32	71.12	62.37	70.29	78.83	67.05	69.05	74.84	74.1	78.48	82.61	-	-	-	-
	60–90	60.03	65.78	75.44	79.28	75.77	81.2	78.55	79.59	82.41	82.7	85.1	86.59	42.03	46.88	55.68	60.08
	40–59	68.26	73.85	79.43	81.69	82.12	86.97	82.09	83.2	83.6	85.23	87.21	87.99	53.29	57.65	63.36	68.18
	20–39	76.07	80.97	82.74	82.04	87.45	89.13	83.46	85.16	84.28	87.04	88.25	89.22	58.61	62.94	66.22	70.93
	<20	83.53	85.8	88.86	89.23	91.8	91.81	86.63	89.82	87.97	91.7	90.22	91.8	64.38	69.23	69.01	75.17
*Enterobacteriaceae*, ESBL+	>90	42.78	21.78	49.24	46.43	48.02	52.61	43.89	46.87	52.83	51.19	57.38	62.43	-	-	-	-
	60–90	47.94	34.67	58.11	52.16	59.47	60.71	55.01	56.55	60.99	61	63.89	66.23	5.91	9.4	13.37	19.66
*Enterobacteriaceae*, ESBL+	40–59	57.78	45.33	63.43	61.45	60.01	64.77	60.24	61.12	62.14	62.99	66.33	68.44	9.27	15.28	20.19	28.22
	20–39	60.49	53.78	66.56	69.03	68.96	69.12	62.35	64.07	63.58	67.35	67.52	70.31	13.22	20.35	23.63	33.03
	<20	61.13	63.38	69.04	69.25	74.3	74.59	65.8	67.46	68	71.41	85.34	87.27	20.67	29.47	29.23	38.93
*Enterobacteriaceae*, ESBL−	>90	58.19	64.65	78.98	71.28	77.9	85.23	74.78	76.07	81.73	81.32	87.54	90.25	-	-	-	-
	60–90	77.46	83.3	87.87	84.8	88.39	91.13	85.13	86.13	89.55	89.54	91.85	93.46	65.87	72.34	83.15	87.09
	40–59	83.45	87.56	89.85	88.33	91.46	93.25	88.3	90.22	91.34	91.96	93.05	93.52	82.46	86.18	91.86	93.2
	20–39	89.28	89.98	91.48	91.26	93.46	94.48	89.14	91.43	92.31	92.95	93.4	94.33	89.21	91.6	93.15	95.17
	<20	92.05	92.82	94.44	94.4	95.69	95.72	92.78	93.31	93.44	94.55	94.79	95.46	92.17	93.63	94.08	95.28
A.baumannii ^b^	>60	22.01	26.76	35.72	29.75	35.17	43.24	52.15	55.35	54.1	57.42	57.52	60.35	3.87	6.87	11.45	19.65
	31–60	71.46	73.48	79.8	78.16	81.24	82.42	53.87	56.68	56.95	57.92	58.57	60.92	25.39	39.35	55.22	66.83
	10–30	81.39	82.08	83.57	85.59	87.71	88.03	55.98	58.69	58.51	60.39	60.6	61.87	62.01	72.13	73.84	82.61
	<10	87.35	87.93	89.05	90.71	90.82	92.13	58.76	60.21	61.59	63.21	63.5	65.75	81.96	90.68	86.43	92.14
*P. aeruginosa*	>90	71.75	76.28	81.9	88.47	86.48	91.16	91	92.79	94.21	93.68	94.95	94.94	-	-	-	-
**Species/CrCL(mL/min)**		**Dosage Regimens (CFR/%)**															
		**PIS**					**PTZ**							**CSL**			
		**3.0 g q8h**	**6.0 g q12h**	**3.0 g q6h**	**4.5 g q8h**	**6.0 g q8h**	**4.5 g q6h**	**4.5 gq12h**	**3.375 g q8h**	**2.25 g q6h**	**4.5 g q8h**	**3.375 g q6h**	**4.5 g q6h**	**3.0 gq12h**	**4.5 gq12h**	**3.0 g q8h**	**4.5 g q8h**
*P. aeruginosa*	60–90	87.14	89.35	91.57	92.82	92.77	94.52	92.35	94.52	95.11	94.83	95.28	95.54	7.15	15.52	23.79	38.44
	40–59	90.8	91.28	94.02	93.61	94.56	95.93	93.46	95.16	95.17	95.44	95.35	95.76	13.22	27.76	41.96	59.01
	20–39	93.58	94.12	94.9	94.47	96.25	96.18	94.11	95.22	95.25	95.51	95.38	95.85	20.81	41.54	52.89	69.97
	<20	95.12	95.32	96.79	96.85	96.86	97.22	96.33	95.32	95.83	96.01	95.54	96.32	42.4	60.99	63.39	76.17

PIS, piperacillin/sulbactam; PTZ, piperacillin/tazobactam; CSL, cefoperazone/sulbactam. The grey background in the table represented CFR achieved above 80% in MCS. a. CSL simulations in patients with different renal function (compared with PIS and PTZ), as detailed in the pharmacokinetic parameters table. b. Pharmacokinetic parameters of sulbactam were used when performing MCS against *A. baumannii.*

**Table 3 antibiotics-12-00363-t003:** Pharmacokinetic parameters used for the Monte Carlo simulations.

	CrCL (mL/min)	No.of Participants	Vd (L)	t_1/2_ (h)	CL (L/h)	CL_R_ (L/h)	PB (%)
Piperacillin	>90	8	14.9 ± 1.6	0.95 ± 5.6	13.5 ± 1.32	10.08 ± 1.2	30
	60–90	8	13.0 ± 1.4	1.10 ± 7.3	9.54 ± 1.14	4.64 ± 0.72	30
	40–59	9	12.5 ± 1.2	1.26 ± 8.5	8.04 ± 0.9	3.48 ± 0.54	30
	20–39	13	12.4 ± 1.0	1.43 ± 8.4	6.84 ± 0.66	2.28 ± 0.36	30
	<20	12	13.1 ± 1.1	1.92 ± 15.0	4.98 ± 0.6	0.96 ± 0.18	30
Cefoperazone	>60	6	11.6 ± 4.8	1.4 ± 0.2	5.66 ± 2.39	0.8 ± 0.08	82–93
	31–60	6	11.1 ± 4.3	1.9 ± 0.9	4.06 ± 1.17	0.82 ± 0.26	82–93
	10–30	6	12.9 ± 4.6	2.5 ± 0.8	3.65 ± 0.81	0.46 ± 0.44	82–93
	<10	6	15.4 ± 5.8	4.0 ± 1.9	2.95 ± 1.2	NA	82–93
Sulbactam	>60	6	30.5 ± 14.5	0.7 ± 0.2	35.57 ± 20.75	10.34 ± 1.94	38
	31–60	6	24.8 ± 8.4	1.6 ± 0.8	11.93 ± 2.75	6.22 ± 1.96	38
	10–30	6	28.5 ± 8.0	4.1 ± 2.9	7.0 ± 4.2	3.97 ± 3.97	38
	<10	6	27.6 ± 8.8	8.4 ± 3.9	2.68 ± 0.98	NA	38

Data are presented as the mean ± SD. Vd, volume of distribution; t_1/2_, half-life; CL, total body clearance; CL_R_, renal clearance; PB, protein binding; CrCL, creatinine clearance.

## Data Availability

All data are applicable in the paper.

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
