# Peer review of "Evaluation of Piperacillin/Sulbactam, Piperacillin/Tazobactam and Cefoperazone/Sulbactam Dosages in Gram-Negative Bacterial Bloodstream Infections by Monte Carlo Simulation"

_antibiotics, 2023, doi:10.3390/antibiotics12020363_

Round 1

Reviewer 1 Report

This is an interesting contribution that should help in the treatment of certain multiply-antibiotic-resistent gram-negative infections.

Author Response

Thanks very much for your confirmation of our manuscript.

Reviewer 2 Report

This paper described the optimal regimens of piperacillin/sulbactam (2:1), piperacillin/tazobactam (8:1), and cefoperazone/sulbactam (2:1) in patients based on renal function by performing antimicrobial susceptibility testing and Monte Carlo simulation. However, there are some concerns in this study.

1.           This manuscript appears to be reporting some optimal regimens by calculating the probability of target attainment, however, the impact is lost due to unclear of pharmacokinetic/pharmacodynamic analysis for each of β-lactams and β-lactamase inhibitors. In Acinetobacter baumannii, which is effected by the β-lactamase inhibitor sulbactam, the susceptibility and optimal dose of each antibacterial agent varies widely. Which did the authors analyze the combination drug or sulbactam, and indicate the dosage in this paper? The authors should analyze and demonstrate clearly the pharmacokinetic/pharmacodynamic of β-lactamase inhibitors, sulbactam and tazobactam.

2.           The ratio of β-lactams to β-lactamase inhibitors and the dosage of each are also vitally important. The authors have to clarify and provide the optimal dosage for each of β-lactams and β-lactamase inhibitors. Piperacillin/sulbactam is not widely used except in China.

Reviewer 3 Report

good work, optimal statical analysis

Author Response

(The authors gave the same response as above.)

Reviewer 4 Report

This article uses Monte Carlo simulations to determine optimal doses of 3 BLBLIs in Gram-negative bacterial bloodstream infections. The popPK models on which the simulations are based are derived from previous studies in healthy volunteers and patients with various degrees of renal function. The complexity of these studies lies in the choice of targets and dose regimen tested for the calculation of PTAs, and my main comments are on these points.

General comments

1) The choice of pharmacokinetic and pharmacodynamic targets is unclear (l.275-276). This target should concern the beta-lactam molecule (piperacillin or cefoperazone). Is that what was done?

If yes, I don't understand Figures 1 and 2: for the same dose regimen and MIC, the PTAs for PIS and PTZ should be the same, but this does not seem to be the case. For example, for a crCL between 20-30 mL/min and a dosage of 4.5g q8h, the PTAs for a MIC of 32 mg/L are about 25% for PIS and almost 100% for PTZ (Figure 1d and 2d).

If a target was also used for the beta-lactam inhibitor, what was it? Did you use a threshold value above which the inhibitor must be to exert its action? This could have been interesting, especially for tazobactam, for which MIC measurements are performed with a fixed value of 4 mg/L. Perhaps it would be desirable to ensure that this concentration is reached in patients?

Furthermore, the selected target (≥40-50%fT > MIC) is based on an older publication. The current consensus for beta-lactams is more of a target of at least >100%fT > MIC.

Please better describe the selected targets and the molecules involved.

2) I also don't understand the choice of dosages tested. The recommended dosages should have been tested.

For PTZ, this is 12 to 18 g / day with a dose every 4 to 6 hours (only 4.5g q6h was tested, not the q4h interval). For a crCL between 20-40 mL/min, 4g q8h (tested) and if crCL < 20 mL/min, 4g q12h (not tested).

For PIS, “recommended doses of 1.5 g or 3.0 g q12h and increased to 6.0 g q12h for the treatment of severe or refractory infections. » (l. 169-170) should also have been tested even if the conclusions even if the results show that higher doses (or shorter intervals) should be used.

Specific comments

1) l. 84 “For strains with an MIC value of 8 mg/L,

This MIC concern piperacillin in case or PIS or PTZ or cefoperazone in case of CSL, that’s it?

2) Table 2 is difficult to read. it may be necessary to use solid lines between germs? And repeat the dose regimen at the top of the second page.

Why the same crCL ranges were not used for A. baumanii ?

3) L.200 “When daily dose of piperacillin in PIS and PTZ administered was the same, the PTA

achieved by PTZ was similar to that of PIS for strains excluding A. baumannii. »

this is in line with my first comment. Does this mean that you used different targets for PIS and PTZ?  And therefore, a particular target of sulbactam?

L.201-204- “For patients with CrCL ≤90 ml/min, the target CFR was achieved when using PTZ at the routinely administered dosing of 4.5 g q8h, whereas for those with CrCL >90 ml/min, an increase to 4.5 g q6h was required to achieve the target CFR.

this is not what I understand by reading table 2

4) Equation 1 seems to be derived from the equation C(t) = C0 x exp (-ke.τ) but can you explain how to reach this equation ? DI is dose interval? f is bioavailability? (why do you use it?)

Round 2

Reviewer 2 Report

The manuscript was revised according to the reviewers’ comments.

Reviewer 4 Report

I thank the authors for their very clear and complete explanations.